# Diagnostic Performance of PET Imaging Using Different Radiopharmaceuticals in Prostate Cancer According to Published Meta-Analyses

**DOI:** 10.3390/cancers12082153

**Published:** 2020-08-04

**Authors:** Salvatore Annunziata, Daniele Antonio Pizzuto, Giorgio Treglia

**Affiliations:** 1Nuclear Medicine Unit, IRCCS Regina Elena National Cancer Institute, 00144 Rome, Italy; salvatoreannunziata@live.it; 2Nuclear Medicine Unit, Fondazione Policlinico Universitario A. Gemelli IRCCS, 00168 Rome, Italy; dapizzuto@gmail.com; 3Department of Nuclear Medicine, Universitaetsspital Zuerich, 8091 Zürich, Switzerland; 4Clinic of Nuclear Medicine, Imaging Institute of Southern Switzerland, Ente Ospedaliero Cantonale, 6500 Bellinzona, Switzerland; 5Academic Education, Research & Innovation Area, General Directorate, Ente Ospedaliero Cantonale, 6500 Bellinzona, Switzerland; 6Department of Nuclear Medicine and Molecular Imaging, Lausanne University Hospital and University of Lausanne, 1011 Lausanne, Switzerland

**Keywords:** PET, choline, PSMA, prostate, meta-analysis, evidence-based, nuclear medicine, imaging

## Abstract

A significant number of meta-analyses reporting data on the diagnostic performance of positron emission tomography (PET) in prostate cancer (PCa) is currently available in the literature. In particular, different PET radiopharmaceuticals were used for this purpose. The aim of this review is to summarize information retrieved by published meta-analyses on this topic. The first step included a systematic search of the literature (last search date: June 2020), screening two databases (PubMed/MEDLINE and Cochrane Library). This combination of key words was used: (A) “PET” OR “positron emission tomography” AND (B) “prostate” OR “prostatic” AND (C) meta-analysis. Only meta-analyses on Positron Emission Tomography/Computed Tomography (PET/CT) or Positron Emission Tomography/Magnetic Resonance (PET/MR) in PCa were selected. We have summarized the diagnostic performance of PET imaging in PCa, taking into account 39 meta-analyses published in the literature. Evidence-based data showed the good diagnostic performance of PET/CT with several radiopharmaceuticals, including prostate-specific membrane antigen (PSMA)-targeted agents, radiolabeled choline, fluciclovine, and fluoride in restaging and staging settings. Less evidence-based data were available for PET/MR with different radiotracers. More prospective multicentric studies and cost-effectiveness analyses are warranted.

## 1. Introduction

Prostate cancer (PCa) is the most frequent type of cancer in men worldwide. PCa patients may be treated with radical prostatectomy or radiotherapy as primary treatments, but they may develop disease recurrence [1].

In the clinical setting of PCa screening, the use of a prostatic-specific antigen (PSA) has led to earlier diagnosis. The first sign of disease recurrence is represented by increasing levels of PSA, known as biochemical recurrence (BR). In early recurrence, potentially curative salvage therapies, such as secondary lymphadenectomy or targeted radiotherapy can be adopted, but both require the disease to be localized [1].

In patients with intermediate- to high-risk PCa and those with BR-PCa, current guidelines recommend disease staging and restaging using conventional imaging, including computed tomography (CT) and magnetic resonance (MR), as well as whole-body bone scintigraphy [1]. In this clinical scenario, morphologic imaging, such as CT, firstly offered a whole-body diagnostic tool both in a staging and restaging setting. At the same time, MR offered a useful tool for the diagnosis of loco-regional disease. About functional imaging, bone scanning with radiolabeled diphosphonates has been widely used for the detection and monitoring of PCa bone metastases: the radiopharmaceutical mimics phosphate and adsorbs to areas of active bone formation, particularly around metastases where osteoblastic activity is prominent.

Unfortunately, anatomical imaging techniques depend solely on morphological features for identifying metastatic disease: as small lesions are frequently missed by morphological imaging techniques, PCa disease can be understaged by using CT or MR. Currently, the general diagnostic performance of morphologic imaging for detecting lymph nodal metastases of PCa remains limited: using histopathology as reference standard, CT and MR showed a sensitivity <50% [1].

About bone scintigraphy, this method may detect lesions not seen by CT in some cases, but it lacks adequate specificity as benign lesions causing increased osteoblastic activity can be mistaken for PCa metastases. In detecting bone metastases at initial diagnosis of PCa, bone scintigraphy presents a relatively low diagnostic yield of 3.5% with PSA ≤ 10 ng/mL, 6.9% with PSA between 10 and 20 ng/mL, and 41.8% with PSA > 20 ng/mL [1]. Lastly, a bone scan only examines the bones and will ignore lymphatic or visceral metastases of PCa [1].

Positron emission tomography (PET) combined with CT (PET/CT) or MR (PET/MR) using several radiopharmaceuticals are emerged as new-generation imaging methods to allow a more accurate and precise staging and restaging of PCa. These whole-body imaging methods combining functional with anatomical information provide an increased diagnostic accuracy in detecting PCa lesions compared to CT, MR, and bone scintigraphy [1].

Different radiotracers for PET imaging have been proposed in recent years, such as radiolabeled choline (Cho) [2,3,4,5,6,7,8,9,10,11,12,13,14,15,16,17,18,19,20], prostate-specific membrane antigen (PSMA)-targeted agents [21,22,23,24,25,26,27,28,29,30,31,32], ^18^F-fluciclovine (FACBC) [8,32,33,34,35,36], ^18^F-fluoride (NaF) [4,37], ^11^C-acetate [11,14,38], and ^18^F-fluorodeoxyglucose (^18^F-FDG) [11,14]. Moreover, different radioisotopes have been proposed to label some of these radiopharmaceuticals, such as to label Cho (e.g., ^18^F, ^11^C) and PSMA-targeted agents (e.g., ^68^Ga, ^18^F, ^64^Cu). Each tracer has different uptake features: radiolabeled Cho and acetate are markers of the cell membrane synthesis; PSMA is a membrane antigen overexpressed in the majority of PCa cells, and it is a target for a variety of urea-derived agents; fluciclovine uptake is related to functional activity of amino acid transporters; fluoride and ^18^F-FDG mark the metabolism of bone matrix and cell glucose, respectively [1]. While PET imaging is usually performed with coregistered CT, hybrid PET/MR was also recently proposed for the management of PCa [5].

To date, several meta-analyses have reported evidence-based data about the diagnostic performance of PET imaging in PCa with different radiotracers [2,3,4,5,6,7,8,9,10,11,12,13,14,15,16,17,18,19,20,21,22,23,24,25,26,27,28,29,30,31,32,33,34,35,36,37,38,39,40]. The aim of this review is to summarize information retrieved by published meta-analyses on this topic.

## 2. Materials and Methods

A comprehensive computer literature search of PubMed/MEDLINE and Cochrane library databases was conducted to find recent published meta-analyses on the diagnostic performance of PET imaging for the diagnosis of PCa.

A search algorithm based on the combination of the following terms was used: (A) “PET” OR “positron emission tomography” AND (B) “prostate” OR “prostatic” AND (C) meta-analysis. The literature search was updated until June 30th, 2020. No language restriction was used. Recent meta-analyses published in the last ten years and investigating the diagnostic performance of PET imaging in PCa were considered eligible for inclusion. Titles and abstracts of the retrieved articles were reviewed, applying the inclusion criteria mentioned above.

For each selected meta-analysis, information was collected about basic study characteristics (authors, year of publication, number of original articles included, number of patients included), and diagnostic performance measures. The main findings of the selected meta-analyses were briefly described.

## 3. Results

Performing a systematic search of the literature (last search date: June 2020) screening two databases (PubMed/MEDLINE and Cochrane Library), 39 meta-analyses were selected [2,3,4,5,6,7,8,9,10,11,12,13,14,15,16,17,18,19,20,21,22,23,24,25,26,27,28,29,30,31,32,33,34,35,36,37,38,39,40], most of them related to PET/CT, and only two focused on PET/MR [5,40]. Main characteristics of the meta-analyses about PET/CT with different tracers are presented in Table 1, Table 2, Table 3, Table 4 and Table 5. The main findings of the selected evidence-based articles are summarized here below taking into account the different types of PET radiopharmaceuticals used for PCa evaluation.

### 3.1. Radiolabeled Choline (Cho) PET/CT

In the first published meta-analyses [19,20], radiolabeled Cho PET and PET/CT showed high sensitivity and specificity for the detection of locoregional and distant metastases in PCa patients with a recurrence of disease (BR-PCa) but low sensitivity and high specificity in the detection of lymph node metastases prior to surgery in intermediate- and high-risk PCa patients. According to the meta-analysis of Evangelista et al. [19], radiolabeled Cho PET and PET/CT provided a pooled sensitivity of 85.6% and a pooled specificity of 92.6% for all sites of disease (prostatic fossa, lymph nodes, and bone) in BR-PCa patients. Conversely, pooled sensitivity and specificity of these methods in detecting PCa lymph nodal metastases prior to surgery were 49.2% and 95%, respectively [20].

These findings were confirmed by Umbehr et al. [18]: in staging PCa patients, on a per-patient basis, radiolabeled Cho PET or PET/CT showed pooled sensitivity and specificity of 84% and 79%, respectively; on a per-lesion basis, these values were 66% and 92%, respectively. In restaging PCa, on a per-patient basis, pooled sensitivity and specificity were 85% and 88%, respectively.

Fanti et al. [13] confirmed the good diagnostic accuracy of ^11^C-Cho PET or PET/CT for the detection of BR-PCa with both pooled sensitivity and specificity equal to 89%.

As demonstrated by von Eyben et al. [12], radiolabeled Cho PET/CT detected metastatic sites in patients with BR-PCa and PSA levels > 1 ng/mL at a clinically relevant level, without a significant impact of types of Cho radiotracers (^11^C-Cho or ^18^F-Cho) and different acquisition protocols on the detection rate.

Evangelista et al. [10] reported that radiolabeled Cho PET/CT has a moderate accuracy for the detection of metastatic lymph nodes in PCa patients who are candidates for salvage lymph node dissection (sensitivity on a patient-based and lesion-based analysis were 85% and 56%, respectively).

Recently, Kim et al. [2] demonstrated that ^18^F-Cho PET/CT has a low sensitivity (57%) and high specificity (94%) for the detection of metastatic lymph nodes in patients with newly diagnosed PCa.

Treglia and colleagues [16] showed a strong relationship between PSA kinetics and the detection rate of radiolabeled Cho PET/CT in BR-PCa. The pooled DR of radiolabeled Cho PET/CT increased when PSA doubling time was ≤ 6 months and when PSA velocity was >1 ng/mL/year, respectively. Therefore, PSA kinetics should be taken into account in the selection of BR-PCa patients who should undergo radiolabeled Cho PET/CT for restaging purpose. At the same time, in another meta-analysis [9] trigger PSA was found as an important risk factor for positive findings of Cho PET/CT: the detection rate of Cho PET/CT for BR-PCa (59%) increased in parallel with rises in PSA values. For trigger PSA > 2 ng/mL the detection rate of ^11^C-Cho PET/CT and ^18^F-Cho PET/CT were 73% and 83%, respectively; the same values were 18% and 39%, respectively, when the trigger PSA was <1 ng/mL.

Beyond the diagnostic performance, radiolabeled Cho PET/CT performed for staging or restaging of PCa led to a change in treatment management in 41% of PCa patients [17].

According to the meta-analysis by Guo et al. [7], pooled sensitivity and specificity of radiolabeled Cho PET/CT in the detection of bone metastases from PCa were 89% and 98%, respectively, on a per-patient based analysis; on a per-lesion basis, the pooled sensitivity and specificity were 91% and 97%, respectively. Shen et al. [15] found higher diagnostic accuracy values for radiolabeled Cho PET/CT and MR compared to bone scintigraphy. To this regard, discordant findings between radiolabeled Cho PET/CT and bone scanning in detecting bone metastases in PCa patients are not negligible [41]. Zhou et al. [4] also demonstrated the superiority of radiolabeled Cho PET/CT compared to bone scintigraphy in this setting; however, radiolabeled Cho PET/CT showed a slightly lower diagnostic accuracy compared to PET/CT with PSMA-targeted agents and ^18^F-NaF for detecting bone metastases of PCa.

Overall, when compared to other PET tracers for restaging of PCa [3,6,8,11,14], no statistically significant differences were reported comparing radiolabeled Cho PET to PET with PSMA-targeted agents or ^18^F-FACBC, even if a superiority of PET with PSMA-targeted agents was demonstrated in detecting PCa lesions at low PSA levels (≤1 ng/mL). On the other hand, a higher diagnostic accuracy of radiolabeled Cho PET compared to ^11^C-acetate PET and ^18^F-FDG PET was reported.

### 3.2. PET/CT with PSMA-Targeted Agents

In the first meta-analysis by Perera et al. [30], trigger PSA predicted the risk of positive PET with ^68^Ga-PSMA-targeted agents in patients with BR-PCa. Pooled data indicated favorable sensitivity (86% on a per-patient basis and 80% on a per-lesion basis) and specificity (86% on a per-patient basis and 97% on a per-lesion basis) of this imaging method. Interestingly, PET/CT with ^68^Ga-PSMA-targeted agents showed clinical relevance to detect sites of recurrence in PCa patients after radical prostatectomy with PSA levels lower than 1.0 ng/mL [29]. According to an updated meta-analysis [24], PET with ^68^Ga-PSMA-targeted agents improved the detection of metastases in BR-PCa, and the detection rate was related to serum PSA values: for PSA level categories ≥ 2, 1–1.99, 0.5–0.99, 0.2–0.49, and 0–0.19 ng/mL, the percentages of positive PET/CT with ^68^Ga-PSMA-targeted agents were 95%, 75%, 59%, 45%, and 33%, respectively. Tan et al. [31] reported that for PSA categories less than 0.5, 0.5 to 0.9, 1 to 1.9, and 2 ng/mL or greater the pooled detection rates of PET/CT with ^68^Ga-PSMA-targeted agents were 44.9%, 61.3%, 78.2%, and 93.9%, respectively. In the meta-analysis by Hope et al. [26] using histopathology a gold standard, the detection rate of PET/CT with ^68^Ga-PSMA-targeted agents was 63% in BR-PCa patients with a PSA less than 2.0 ng/mL and 94% in BR-PCa patients with a PSA greater than 2.0 ng/mL. Notably, shorter PSA kinetics may be a predictor of positivity of PET with PSMA-targeted agents in patients with BR-PCa [25].

According to the meta-analysis of Kimura et al. [21], PET/CT with PSMA-targeted agents performed before salvage node dissection in BR-PCa yielded high sensitivity and specificity. The pooled sensitivity using lesion-based and field-based analyses were 84% and 82%, respectively. The pooled specificity using lesion-based and field-based analyses were 97% and 95%, respectively.

In a recent meta-analysis [22], PET/CT with ^18^F-PSMA-targeted agents showed a good detection rate in BR-PCa. The detection rate of PET/CT with ^18^F-PSMA-targeted agents was influenced by serum PSA with lower values in PCa patients with PSA < 0.5 ng/mL (49%) compared to those with PSA ≥ 0.5 ng/mL (86%).

In a head-to-head comparative meta-analysis [3], PET/CT with PSMA-targeted agents was superior in detecting PCa lesions only at low PSA levels (≤1 ng/mL) when compared to radiolabeled Cho PET/CT in BR-PCa; for PSA ≤ 1 ng/mL, the detection rate of PET/CT with radiolabeled Cho and PSMA-targeted agents were 27% and 54%, respectively. Conversely, in BR-PCa patients with PSA > 1 ng/mL, a statistically significant difference in detection rate among these two methods has not been reported by evidence-based manuscripts [3,6]. In a comparative analysis among PET/CT with PSMA-targeted agents and ^18^F-FACBC in BR-PCa with PSA levels < 2 ng/mL [32], PET/CT with PSMA-targeted agents had a significantly higher detection rate (80%) compared to PET/CT with ^18^F-FACBC (62%) for PSA levels of 1.0–1.9 ng/mL.

In a meta-analysis about the staging setting, PET/CT with PSMA-targeted agents showed a moderate sensitivity (71%) and high specificity (95%) for the detection of metastatic lymph nodes in patients with newly diagnosed intermediate to high-risk PCa [27]. In another meta-analysis using pathology as a gold standard [26], PET/CT with ^68^Ga-PSMA-targeted agents at initial PCa staging demonstrated a sensitivity and specificity of 74% and 96%, respectively. In staging intermediate to high-risk PCa, PET/CT with ^68^Ga-PSMA-targeted agents had a higher sensitivity compared to MR (65% versus 41%, respectively) and similar specificity (94% versus 92%, respectively) in detecting lymph nodal metastases [23].

Compared to other PET methods, PET/CT with PSMA-targeted agents had the highest patient-based sensitivity (97%) and specificity (100%) in detecting bone metastases [4].

According to the meta-analysis of Han et al. [28] focused on the impact of PET/CT with ^68^Ga-PSMA-targeted agents in PCa patients, this imaging method altered the management in about half of PCa patients.

### 3.3. ^18^F-FACBC (fluciclovine) PET/CT

In some meta-analyses, diagnostic accuracy of ^18^F-FACBC PET/CT was reported as moderate: higher in the staging than restaging setting [8,32,33,34,35,36]. In the first meta-analysis by Ren et al. [36], ^18^F-FACBC PET/CT demonstrated a pooled sensitivity and pooled specificity of 87% and 66%, respectively, in detecting BR-PCa. In another study, ^18^F-FACBC PET/CT had lower accuracy values than radiolabeled Cho PET in a restaging setting: the pooled sensitivity was similar (80.9% for radiolabeled Cho PET/CT and 79.7% for ^18^F-FACBC PET/CT, respectively), but the pooled specificity was higher for radiolabeled Cho PET/CT (84.1%) compared to ^18^F-FACBC PET/CT (61.9%) [8]. In a comparative analysis among PET/CT with ^18^F-FACBC and PSMA-targeted agents in BR-PCa with PSA levels < 2 ng/mL [32], pooled detection rates for PET/CT with ^18^F-FACBC were 37% for a PSA level less than 0.5 ng/mL, 48% for a PSA level of 0.5–0.9 ng/mL, and 62% for a PSA level of 1.0–1.9 ng/mL, with significant lower detection rate of ^18^F-FACBC PET/CT compared to PET/CT with PSMA-targeted agents in the last subgroup of patients.

Based on the meta-analysis by Kim et al. [35], the pooled sensitivity for ^18^F-FACBC PET or PET/CT for diagnosis of primary PCa was 87% and the pooled specificity 84%. For lymph nodal staging, the pooled sensitivity was 56% and the pooled specificity was 98%. For detection of BR-PCa, the pooled sensitivity was 79% and the pooled specificity was 69%.

Another two recent meta-analyses [33,34] confirmed the good sensitivity (86–88%) and moderate specificity (73–76%) of ^18^F-FACBC PET/CT in staging and restaging PCa.

### 3.4. ^11^C-Acetate PET/CT

^11^C-acetate PET/CT showed intermediate or low diagnostic accuracy values in PCa according to available meta-analyses, both in staging and restaging settings [11,14,38]. According to the meta-analysis of Mohsen et al. [38], for the evaluation of primary tumors, ^11^C-acetate PET showed moderate pooled sensitivity (75.1%) and specificity (75.8%) [38]. For detection of recurrence, pooled sensitivity was 64% and pooled specificity was 93%. Sensitivity for recurrence detection was higher in PCa patients with serum PSA values at relapse > 1 ng/mL [38].

Overall, compared to radiolabeled Cho PET/CT, ^11^C-acetate PET/CT seems to have similar sensitivity in localizing sites of relapse in BR-PCa. The complete lack of urinary excretion of ^11^C-acetate offers a distinct advantage over radiolabeled Cho in image interpretation [42,43].

### 3.5. ^18^F-NaF (Fluoride) PET/CT

Two meta-analyses showed high-accuracy values of ^18^F-NaF PET/CT in the detection of bone metastases of patients with PCa [4,37]. Sheikhbahaei et al. reported a pooled sensitivity and specificity of 98% and 90%, respectively, with better overall accuracy of ^18^F-NaF PET/CT compared to bone scintigraphy and MR [37]. Even if PET/CT with the other radiotracers mentioned above is also able to detect bone metastases, potentially with even greater sensitivity than ^18^F-NaF PET/CT (e.g., PET/CT with PSMA-targeted agents may detect lesions in bone marrow even before associated lesions of the bone matrix), the use of ^18^F-NaF PET/CT in the detection of bone metastases in PCa generally represents a clinical application in patients with far more advanced disease than biochemical recurrence [4,37].

### 3.6. ^18^F-FDG PET/CT

As demonstrated by some evidence-based articles, ^18^F-FDG PET/CT showed lower accuracy values than other PET tracers in detecting PCa [11,14].

In a meta-analysis by Bertagna et al. [39], prostate incidental uptake was observed in about 2% of ^18^F-FDG PET/CT scans performed in male patients. The pooled risk of malignancy verified by biopsy was about 60%. Peripheric site but not presence or absence of calcification was a predictor of malignancy. Therefore, whenever an incidental ^18^F-FDG uptake is detected in the prostate, further investigation should be warranted to exclude malignancy, in particular when located in the peripheric site of the prostate gland [39].

### 3.7. PET/MR with Different Radiotracers

A meta-analysis evaluated the accuracy of hybrid PET/MR imaging in PCa by using different radiopharmaceuticals, showing higher sensitivity of PET/MR compared to multiparametric MR for the diagnosis of primary PCa [5]. Another recent meta-analysis focused on PET/MR with ^68^Ga-PSMA-targeted agents [40] found that this method is likely effective in the diagnosis of primary PCa, and its diagnostic accuracy BR-PCa was positively correlated to serum PSA levels: The pooled sensitivity and specificity in detecting primary PCa were 83% and 81%, respectively; in BR-PCa, the pooled detection rate was 76%. For PSA level categories 0−0.2 ng/mL, 0.2−1 ng/mL, 1−2 ng/mL, and more than 2 ng/mL, the pooled detection rates were 38%, 67%, 74%, and 95%, respectively.

Notably, current combined PET/MR cameras do not employ state-of-the-art detectors and reconstruction technologies that would be typically present in current clinical PET/CT scanners, where the resulting improvements in resolution and sensitivity are clearly relevant to the problem of finding small lesions.

## 4. Discussion

To date, several meta-analyses reported data about the diagnostic performance of PET imaging in PCa with different radiotracers [2,3,4,5,6,7,8,9,10,11,12,13,14,15,16,17,18,19,20,21,22,23,24,25,26,27,28,29,30,31,32,33,34,35,36,37,38,39,40]. Since the first evidence-based papers were published, radiolabeled Cho PET/CT showed high diagnostic accuracy in a restaging setting of patients with BR-PCa [13,19], particularly for the detection of locoregional and distant metastases. Subsequently, the choice between the ^18^F or ^11^C-Cho radiotracers and different acquisition protocols showed no significant impact on diagnostic accuracy [12], being useful for several clinical purposes, such as the guide of salvage lymph node dissection [10]. Nevertheless, PSA value quickly emerged as a critical aspect in patients eligible for radiolabeled Cho PET/CT, since this method showed higher detection rate for trigger PSA > 2 ng/mL [9,16]. According to those evidences, radiolabeled Cho PET/CT is widely used in the daily routine for the restaging of BR-PCa in patients with intermediate and high PSA values [1].

At the same time, the use of radiolabeled Cho for staging purpose before first-line treatment is still debated. Firstly, Cho PET/CT provided low sensitivity but high specificity in the detection of lymph node metastases [2,20]. Conversely, other studies found good overall accuracy of Cho PET/CT in this setting [17], and also when compared to other functional and morphological imaging tools [11,14]. The best diagnostic performance of Cho PET/CT for staging purpose was shown for the detection of bone metastases [7,15]. For these reasons, radiolabeled Cho is not extensively used for staging purposes in PCa patients: more evidences are needed in order to promote the routine use of Cho PET in this setting [1].

More recently, PSMA-targeted radiotracers emerged in order to overcome the limitations of Cho PET in the diagnosis of early recurrence of PCa. Indeed, several studies showed high diagnostic accuracy of PET/CT with PSMA-targeted agents in PCa both for restaging and staging purpose. One of the advantages of receptor-based radiopharmaceuticals, such as PSMA-targeted agents, compared to metabolic tracers (i.e., radiolabeled Cho or FACBC) is the higher target-to-background ratio resulting in higher sensitivity and inter-reader agreement [44]. Since the first published meta-analyses, compared to radiolabeled Cho PET/CT, PET/CT with ^68^Ga-PSMA-targeted agents showed a higher detection rate in a restaging setting in patients with serum PSA levels < 1.0 ng/mL [22,24,25,30,31]. Using histopathology as the gold standard, the detection rate of PET/CT with ^68^Ga-PSMA-targeted agents was 63% in BR-PCa patients with a PSA < 2 ng/mL and 94% in BR-PCa patients with a PSA > 2.0 ng/mL [26]. Moreover, PET/CT with ^18^F-PSMA-targeted agents recently showed a good detection rate in a restaging setting: the pooled DR was 86% for PSA ≥ 0.5 ng/mL and 49% for PSA < 0.5 ng/mL [22]. Furthermore, PET/CT with PSMA-targeted agents showed good accuracy values in some meta-analyses about staging setting before first-line treatment [6,23,26,27], with particular regard to bone metastases [4]. All the meta-analyses here discussed showed an increasing interest about the role of PET/CT with PSMA-targeted agents due to their good diagnostic performance in several clinical settings of PCa [1]. PSMA-targeted agents are also an excellent theragnostic agent allowing the detection of PCa lesions by PET/CT imaging and, subsequently, the irradiation of metastatic sites with beta or alpha particle emitters [44]. Despite these clear evidences, PSMA-targeted agents are still classified as experimental radiopharmaceuticals in several countries at the moment, and this has an influence on the use of these PET radiopharmaceuticals in the clinical practice.

Finally, less evidence-based data are available about other PET radiopharmaceuticals. ^18^F-FDG PET/CT could show incidental radiopharmaceutical uptake in the prostate addressing further investigation [39], but low accuracy values emerged in staging PCa [11,14]. ^11^C-acetate PET/CT showed intermediate or low-accuracy values both in PCa staging and restaging setting [11,14,38]. ^18^F-FACBC PET/CT showed good accuracy values, with better performance in the staging than in restaging setting [8,32,33,34,35,36]. High accuracy values of ^18^F-NaF PET/CT emerged in the detection of bone metastases of patients with PCa [4,37]. Few evidence-based data were found for hybrid PET/MR with different radiotracers [5,40].

Some limitations of PET imaging with different radiopharmaceuticals should be briefly discussed. In particular, the detection rate of PCa lesions with the several PET radiopharmaceuticals described seems to be strictly related to serum PSA values and PSA kinetics. Therefore, false negative cases are more common in PCa patients with lower PSA serum values or slower PSA kinetics. Moreover, false positive findings due to radiopharmaceutical uptake in other tumors or benign conditions should be taken into account [45].

Some limitations of the selected meta-analyses should be discussed too, because they could hamper the achievement of definitive conclusions on the diagnostic performance of PET with several radiopharmaceuticals in PCa. First of all, in some meta-analyses a limited number of articles were included, thus reducing the statistical power of the analysis. In several meta-analyses, a significant statistical heterogeneity among the included studies in terms of diagnostic accuracy outcomes has been identified. This heterogeneity may be explained, for instance, by differences between the characteristics of PCa patients, methodological aspects, quality of the studies, and PET interpretation criteria [46,47]. Many of the studies included in the selected meta-analyses are single-center and retrospective series, limiting their applicability to a broader setting. The lack of a reliable “gold standard” as histopathology for the diagnosis of PCa in some articles could be another limitation of the described meta-analyses (verification bias), causing a possible overestimation of diagnostic test performance. Lastly, reporting bias (such as publication bias when studies reporting significant findings are more likely to be published than those reporting nonsignificant results) cannot be excluded in some selected meta-analyses [46,47].

Our review was focused on collecting data about the diagnostic performance only. We have to underline that diagnostic performance is not a measure of clinical effectiveness, and good diagnostic accuracy of an imaging method does not necessarily result in improved patient outcomes. Therefore, other factors beyond the diagnostic performance should influence the choice of an imaging modality in patients with PCa both at staging and restaging (e.g., availability, safety, legal, organization and economic aspects, and cost-effectiveness). Therefore, recommendations on the use of PET/CT or PET/MR with different radiopharmaceuticals in PCa should not be provided, taking into account only information on the diagnostic performance [48].

According to recent international guidelines, when conventional imaging is negative or equivocal in PCa patients with a high risk of metastatic disease, PET/CT or PET/MRI with different radiopharmaceuticals may add clinical benefit, although prospective data are limited [1]. In restaging PCa patients with negative conventional imaging for men for whom salvage therapy is contemplated, PET/CT or PET/MRI with different radiopharmaceuticals (including PSMA-targeted agents or radiolabeled Cho or ^18^F-FACBC or ^18^F-NaF) is recommended as they have superior disease detection performance characteristics and may alter patient management [1].

Overall, based on current evidence-based data, we would like to recommend more prospective multicentric studies and cost-effectiveness analyses for all PET tracers in order to clarify and strengthen their use in the clinical practice.

## 5. Conclusions

Evidence-based data showed the good diagnostic performance of PET imaging with several radiopharmaceuticals in different PCa clinical settings, including staging and restaging. In BR-PCa patients with low serum PSA values, PET with PSMA-targeted agents seems to provide a higher detection rate compared to PET with other radiopharmaceuticals. More prospective multicentric studies and cost-effectiveness analyses are warranted.

## Figures and Tables

**Table 1 cancers-12-02153-t001:** Meta-analyses on radiolabeled choline PET/CT in prostate cancer.

Authors	Year	Articles	Patients	Topic	Pooled Sensitivity	Pooled Specificity	Pooled LR+	Pooled LR−	Pooled DOR	Pooled DR
Kim et al. [2]	2019	7	627	Staging	0.57	0.94	10.2	0.46	22	NA
Treglia et al. [3]	2019	5	257	Restaging	NA	NA	NA	NA	NA	0.56
Zhou et al. [4]	2019	11	NA	Detection of bone metastases	0.87	0.99	NA	NA	504	NA
Lin et al. [6]	2019	16	2122	Staging/restaging	0.93	0.83	4.98	0.10	68.27	NA
Sathianathen et al. [8]	2019	16	NA	Restaging	0.81	0.84	5.4	0.24	25.2	0.62
Guo et al. [7]	2018	14	NA	Detection of bone metastases	0.89	0.98	40.4	0.12	344	NA
Wei et al. [9]	2018	44	NA	Restaging	0.82	0.92	6.61	0.20	38.55	0.59
Evangelista et al. [10]	2016	9	NA	Restaging	0.85	0.33	1.21	0.46	2.83	NA
Liu et al. [11]	2016	46	NA	Staging/restaging	0.76–0.83	0.82–0.93	4.5–11.7	0.21–0.26	22–46	NA
Von Eyben et al. [12]	2016	18	2219	Restaging	NA	NA	NA	NA	NA	0.55
Fanti et al. [13]	2016	18	2126	Restaging	0.89	0.89	NA	NA	NA	0.62
Ouyang et al. [14]	2016	46	NA	Staging	0.73–0.78	0.79–0.90	NA	NA	NA	NA
Shen et al. [15]	2014	9	NA	Detection of bone metastases	0.91	0.99	NA	NA	150.70	NA
Treglia et al. [16]	2014	14	NA	Restaging	NA	NA	NA	NA	NA	0.58
von Eyben et al. [17]	2014	47	3167	Staging/restaging	0.59	0.92	6.86	0.45	19.17	NA
Umbehr et al. [18]	2013	29	1843	Staging/restaging	0.84/0.85	0.79/0.88	4.02/7.06	0.2/0.17	20.4/41.4	NA
Evangelista et al. [19]	2013	19	1555	Restaging	0.86	0.93	NA	NA	62.12	NA
Evangelista et al. [20]	2013	10	441	Staging	0.49	0.95	8.35	0.55	18.99	NA

Legend: NA = not available; LR+ = positive likelihood ratio; LR- = negative likelihood ratio; DOR = diagnostic odds ratio; DR = detection rate.

**Table 2 cancers-12-02153-t002:** Meta-analyses on PET/CT with PSMA-targeted agents in prostate cancer.

Authors	Year	Articles	Patients	Topic	Pooled Sensitivity	Pooled Specificity	Pooled LR+	Pooled LR−	Pooled DOR	Pooled DR
Tan et al. [32]	2020	38	3217	Restaging	NA	NA	NA	NA	NA	NA (*)
Kimura et al. [21]	2020	9	NA	Restaging	0.84	0.97	30.3	0.16	189	NA
Wu et al. [23]	2020	13	1597	Staging	0.65	0.94	10.6	0.37	29	NA
Perera et al. [24]	2020	37	4790	Staging/restaging	0.77	0.97	NA	NA	NA	NA
Tan et al. [31]	2019	43	5113	Restaging	NA	NA	NA	NA	NA	0.70
Treglia et al. [3]	2019	5	257	Restaging	NA	NA	NA	NA	NA	0.78
Zhou et al. [4]	2019	6	NA	Detection of bone metastases	0.97	1.00	NA	NA	NA	NA
Treglia et al. [22]	2019	6	645	Restaging	NA	NA	NA	NA	NA	0.81
Lin et al. [6]	2019	13	652	Staging/restaging	0.92	0.94	7.91	0.14	79.04	NA
Pereira Mestre et al. [25]	2019	8	NA	Restaging	NA	NA	NA	NA	NA	0.72
Hope et al. [26]	2019	29	NA	Staging/restaging	0.74/0.99	0.96/0.76	NA	NA	NA	NA
Kim et al. [27]	2019	6	298	Staging	0.71	0.95	15.6	0.30	51	NA
Han et al. [28]	2018	15	1163	Impact on management	NA	NA	NA	NA	NA	NA
Von Eyben et al. [29]	2018	15	1256	Staging/restaging	0.61–0.7/0.87–0.93	0.84–0.97/0.93–1	NA	NA	NA	0.74/0.81
Perera et al. [30]	2016	16	1309	Staging/restaging	0.86	0.86	NA	NA	NA	0.40/0.76

Legend: NA = not available; LR+ = positive likelihood ratio; LR− = negative likelihood ratio; DOR = diagnostic odds ratio; DR = detection rate; (*) = only available for PSA < 2 ng/mL.

**Table 3 cancers-12-02153-t003:** Meta-analyses on radiolabeled fluciclovine (FACBC) PET/CT in prostate cancer.

Authors	Year	Articles	Patients	Topic	Pooled Sensitivity	Pooled Specificity	Pooled LR+	Pooled LR−	Pooled DOR	Pooled DR
Tan et al. [32]	2020	6	482	Restaging	NA	NA	NA	NA	NA	NA (*)
Bin et al. [33]	2020	9	363	Staging/restaging	0.88	0.73	3.3	0.17	20	NA
Laudicella et al. [34]	2019	9	NA	Staging/restaging	0.86	0.76	4.5	0.34	16.4	NA
Kim et al. [35]	2019	13	563	Staging/restaging	0.56–0.87/0.79	0.84–0.98/0.69	5.3–19.3/2.5	0.16–0.48/0.3	34–44/9	NA
Sathianathen et al. [8]	2019	5	NA	Restaging	0.80	0.62	2.1	0.36	8	0.59
Ren et al. [36]	2016	6	251	Restaging	0.87	0.66	NA	NA	NA	NA

Legend: NA = not available; LR+ = positive likelihood ratio; LR− = negative likelihood ratio; DOR = diagnostic odds ratio; DR = detection rate; (*) = only available for PSA < 2 ng/mL

**Table 4 cancers-12-02153-t004:** Meta-analyses on radiolabeled acetate PET/CT in prostate cancer.

Authors	Year	Articles	Patients	Topic	Pooled Sensitivity	Pooled Specificity	Pooled LR+	Pooled LR−	Pooled DOR	Pooled DR
Liu et al. [11]	2016	5	NA	Staging/restaging	0.79	0.59	1.90	0.35	6	NA
Ouyang et al. [14]	2016	5	NA	Staging	0.79	0.59	NA	NA	NA	NA
Mohsen et al. [38]	2013	23	NA	Staging/restaging	0.75/0.64	0.76/0.93	1.8	0.45	3.9	NA

Legend: NA = not available; LR+ = positive likelihood ratio; LR− = negative likelihood ratio; DOR = diagnostic odds ratio; DR = detection rate.

**Table 5 cancers-12-02153-t005:** Meta-analyses on fluoride PET/CT in prostate cancer.

Authors	Year	Articles	Patients	Topic	Pooled Sensitivity	Pooled Specificity	Pooled LR+	Pooled LR−	Pooled DOR	Pooled DR
Zhou et al. [4]	2019	7	NA	Detection of bone metastases	0.96	0.97	NA	NA	674	NA
Sheikhbahaei et al. [37]	2019	12	507	Detection of bone metastases	0.98	0.90	6.6	0.07	123.2	NA

Legend: NA = not available; LR+ = positive likelihood ratio; LR− = negative likelihood ratio; DOR = diagnostic odds ratio; DR = detection rate.

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
