# Peer review of "Diagnostic Performance of PET Imaging Using Different Radiopharmaceuticals in Prostate Cancer According to Published Meta-Analyses"

_cancers, 2020, doi:10.3390/cancers12082153_

Round 1

Reviewer 1 Report

Lines 32, 53, and 58.  The authors’ use of the “PSMA” acronym stands to be confusing.  PSMA cannot be used to both represent “prostate specific membrane antigen” (the TARGET being imaged) AND the agent/radiopharmaceutical directed to that target. In line 32, it should instead refer to “PSMA-targeted agents” (since the comparison is to another agent, choline).  Line 53 should similarly refer to “PSMA-targeted agents”, not PSMA. Line 58, says “PSMA is a ligand”, when in fact PSMA is a target (for a variety of urea-derived agents).

Line 59.  “Despite” should perhaps instead read “While”.

Lines 94-119.  The authors have focused on the literature around the utility of PSMA-targeted agents in biochemical recurrence patients with fairly low serum PSA values.  It may be useful to also indicate that PSMA-targeted PET has quite good reliability for finding recurrence sites, if serum PSA levels have reached the 2-10 ng/mL range (and above).

Lines 120-126.  While perhaps a correct summary of meta-analysis findings for 11C-acetate, there is concern that the description does not capture findings from head-to-head studies concluding that choline and acetate are equally effective in localizing sites of biochemical recurrence.  In fact, if equal in sensitivity and specificity, the complete lack of urinary excretion of acetate offers a distinct advantage over choline in image interpretation.

  1. Kotzerke J, Volkmer BG, Glatting G, et al. Intraindividual comparison of [11C] acetate and [11C]choline PET for detection of metastases of prostate cancer.Nuklearmedizin. 2003;42:25–30.
  1. Jadvar H. Prostate Cancer: PET with 18F-FDG, 18F- or 11C-Acetate, and 18F or 11C-Choline. J Nucl Med 2011: 52:81-89.

Lines 136-139. It is perhaps worth emphasizing that the use of F-18 fluoride PET in detection of bone metastases generally represents a clinical application in patients with far more advanced disease than the “biochemical recurrence” cases previously discussed.  (PSMA-targeted PET will also detect bone metastases, potentially with even greater sensitivity than fluoride, since the PSMA-target agents will detect lesions in bone marrow even before associated lesion fracture of the bone matrix.)

Line 212.  A caveat relevant to the statement about limited data being available with PET/MR.  Current combined PET/MR cameras do not employ state-of-the-art detector and reconstruction technologies that would be typically be present in current clinical PET/CT scanners, where the resulting improvements in resolution and sensitivity are clearly relevant to the problem of finding small lesions (as may be required in early imaging of patients with biochemical recurrence). Therefore, PET/MR performance may not be accurately predicted by simple extrapolation from PET/CT findings.

Author Response

Response to Reviewer #1

1) Lines 32, 53, and 58. The authors’ use of the “PSMA” acronym stands to be confusing. PSMA cannot be used to both represent “prostate specific membrane antigen” (the TARGET being imaged) AND the agent/radiopharmaceutical directed to that target. In line 32, it should instead refer to “PSMA-targeted agents” (since the comparison is to another agent, choline). Line 53 should similarly refer to “PSMA-targeted agents”, not PSMA. Line 58, says “PSMA is a ligand”, when in fact PSMA is a target (for a variety of urea-derived agents).

Response: We thank the Reviewer for these useful suggestions. We have changed the term PSMA according to the reviewer’s suggestion.

2) Line 59. “Despite” should perhaps instead read “While”.

Response: We have changed “despite” with “while” according to the reviewer’s suggestion.

3) Lines 94-119. The authors have focused on the literature around the utility of PSMA-targeted agents in biochemical recurrence patients with fairly low serum PSA values. It may be useful to also indicate that PSMA-targeted PET has quite good reliability for finding recurrence sites, if serum PSA levels have reached the 2-10 ng/mL range (and above).

Response: We have added more data about the detection rate values of PET/CT with PSMA-targeted agents in biochemical recurrent prostate cancer adding detection rate values for different categories of serum PSA levels.

4) Lines 120-126. While perhaps a correct summary of meta-analysis findings for 11C-acetate, there is concern that the description does not capture findings from head-to-head studies concluding that choline and acetate are equally effective in localizing sites of biochemical recurrence. In fact, if equal in sensitivity and specificity, the complete lack of urinary excretion of acetate offers a distinct advantage over choline in image interpretation.

Kotzerke J, Volkmer BG, Glatting G, et al. Intraindividual comparison of [11C] acetate and [11C]choline PET for detection of metastases of prostate cancer. Nuklearmedizin. 2003;42:25–30.
Jadvar H. Prostate Cancer: PET with 18F-FDG, 18F- or 11C-Acetate, and 18F or 11C-Choline. J Nucl Med 2011: 52:81-89.

Response: According to the reviewer’s suggestion, we have added the following statement in the paragraph related to 11C-acetate PET: “Overall, compared to radiolabeled Cho PET/CT, 11C-acetate PET/CT seems to have similar sensitivity in localizing sites of relapse in BR-PCa. The complete lack of urinary excretion of acetate offers a distinct advantage over radiolabeled Cho in image interpretation.” We have also added the references suggested by the Reviewer.

5) Lines 136-139. It is perhaps worth emphasizing that the use of F-18 fluoride PET in detection of bone metastases generally represents a clinical application in patients with far more advanced disease than the “biochemical recurrence” cases previously discussed.  (PSMA-targeted PET will also detect bone metastases, potentially with even greater sensitivity than fluoride, since the PSMA-target agents will detect lesions in bone marrow even before associated lesion fracture of the bone matrix.)

Response: We have added the statements suggested by the Reviewer in the paragraph about fluoride PET/CT.

6) Line 212. A caveat relevant to the statement about limited data being available with PET/MR.  Current combined PET/MR cameras do not employ state-of-the-art detector and reconstruction technologies that would be typically be present in current clinical PET/CT scanners, where the resulting improvements in resolution and sensitivity are clearly relevant to the problem of finding small lesions (as may be required in early imaging of patients with biochemical recurrence). Therefore, PET/MR performance may not be accurately predicted by simple extrapolation from PET/CT findings.

Response: We have modified the paragraph on PET/MR adding the suggestions of the Reviewer.

Reviewer 2 Report

This review sought to sum up the published evidence-based data about diagnostic performance of PET imaging in newly diagnosed and recurrent prostate cancer with different radiopharmaceuticals like [11C]choline, [68Ga]/[18F]PSMA, [11C]acetate, [18F]fluciclovine, [18F]fluorid and [18F]FDG. Therefore, a computer-based literature search of meta-analyses published in PubMed/MEDLINE and Cochrane databases until March 2020 was performed. The authors concluded a good diagnostic performance of PET imaging in prostate cancer independent of the clinical setting and applied radiopharmaceutical.

Broad comments

The topic of the review is meaningful since PET imaging is broadly used in diagnosis and staging of prostate cancer nowadays. Depending on availability and requirements for cost reimbursement different radiopharmaceuticals are used. Highlighting the evidence for the different radiopharmaceuticals in different clinical settings might be helpful to select the ideal tracer.

Nevertheless, major drawbacks have to be mentioned:

The most important question in the daily routine is the sensitivity and specificity of PET imaging depending on the PSA level (PSA cutoffs that had positive imaging results). This is extensively investigated in most of the cited articles but not adequate addressed in the review at all (also missing in the tables).  See, by analogy Hope TA et al. J Nucl Med, 2019.

Furthermore, the authors described the diagnostic yield for each radiopharmaceutical in a too general way (“high sensitivity”; “good diagnostic accuracy”; “excellent diagnostic performance”). In this case, raw numbers are required to assess the diagnostic accuracy.

Introduction:

  • The authors should explain the advantages of PET in terms of diagnostic accuracy over conventional imaging.
  • Please provide data for the diagnostic yield of common imaging techniques (bone scan, CT, MRT) in the primary and recurrent setting.
  • Make sure that so far international guidelines (like EAU guideline) only recommend PET imaging in patients with biochemical recurrence.

Discussion:

  • The results should be discussed in a proper way taking into account the limitations of PET imaging in general (accumulation in different malignant and benign conditions; rare histopathological confirmation).
  • Provide sufficient data especially for diagnostic yield of PSMA-PET/CT (discussion line 188 – 202).

Conclusions:

  • The authors should mention the best performing tracer in each setting.

Specific comments

  • Page 1, line 38: Up to 40% of patients treated with radical prostatectomy as primary treatment develop a recurrence over a 10-year period. Please provide reference.
  • Page 2, line 47: About functional imaging, bone scan was widely used for the detection and monitoring of bone metastases from PCa. Please correct.
  • The tables should be in line with the detailed list of the different radiopharmaceuticals in the results
  • Discussion: Avoid word repetitions (e.g. recent)
  • Discussion, line 197: Interestingly, a recent study underlined the impact of 68Ga-PSMA in clinical management of PCa patients [28]. Can be deleted.

Author Response

Response to Reviewer #2

1) The most important question in the daily routine is the sensitivity and specificity of PET imaging depending on the PSA level (PSA cutoffs that had positive imaging results). This is extensively investigated in most of the cited articles but not adequate addressed in the review at all (also missing in the tables). See, by analogy Hope TA et al. J Nucl Med, 2019.

Response: According to the Reviewer’s comment, we have added more data in the text about the detection rate of PET with different radiopharmaceuticals taking into account different serum PSA cut-off values.

2) Furthermore, the authors described the diagnostic yield for each radiopharmaceutical in a too general way (“high sensitivity”; “good diagnostic accuracy”; “excellent diagnostic performance”). In this case, raw numbers are required to assess the diagnostic accuracy.

Response: According to the Reviewer’s comment, we have extensively revised the text adding more quantitative measures about the diagnostic performance of PET with different radiopharmaceuticals.

3) The authors should explain the advantages of PET in terms of diagnostic accuracy over conventional imaging.

Response: We have added a statement in the introduction explaining the advantages of PET in terms of diagnostic accuracy compared to conventional imaging.

4) Please provide data for the diagnostic yield of common imaging techniques (bone scan, CT, MR) in the primary and recurrent setting.

Response: We have added more data in the introduction about the diagnostic yield of conventional imaging methods in PCa.

5) Make sure that so far international guidelines (like EAU guideline) only recommend PET imaging in patients with biochemical recurrence.

Response: We have added more information on recommendations by international guidelines on the use of PET imaging in prostate cancer at the end of the discussion section.

6) The results should be discussed in a proper way taking into account the limitations of PET imaging in general (accumulation in different malignant and benign conditions; rare histopathological confirmation).

Response: We have added more information on limitations of PET imaging and also limitations of the included meta-analyses in the discussion section.

7) Provide sufficient data especially for diagnostic yield of PSMA-PET/CT

Response: We have added more information on the diagnostic yield of PSMA-PET/CT even in the discussion.

8) Conclusions: The authors should mention the best performing tracer in each setting.

Response: We have mentioned the best performing tracer in the conclusions.

9) Page 1, line 38: Up to 40% of patients treated with radical prostatectomy as primary treatment develop a recurrence over a 10-year period. Please provide reference.

Response: We have modified this statement deleting percentages.

10) Page 2, line 47: About functional imaging, bone scan was widely used for the detection and monitoring of bone metastases from PCa. Please correct.

Response: We have corrected the statement.

11) The tables should be in line with the detailed list of the different radiopharmaceuticals in the results

Response: The tables are now in line with the description of different radiopharmaceutical in the main text.

12) Discussion: Avoid word repetitions (e.g. recent)

Response: We have avoided word repetitions in the main text of the discussion.

13) Discussion, line 197: Interestingly, a recent study underlined the impact of 68Ga-PSMA in clinical management of PCa patients [28]. Can be deleted.

Response: We have deleted the statement.

Round 2

Reviewer 2 Report

Major drawbacks were sufficiently edited  and the authors responded to all comments point-by-point.

After revision quality of the paper has been significantly
improved and can now be accepted for publication.